# Access to Water and Sanitation Infrastructures for Primary Schoolchildren in the South-Central Part of Côte d’Ivoire

**DOI:** 10.3390/ijerph18168863

**Published:** 2021-08-23

**Authors:** Parfait K. Kouamé, Anaïs Galli, Maryna Peter, Georg Loss, Diarassouba Wassa, Bassirou Bonfoh, Jürg Utzinger, Mirko S. Winkler

**Affiliations:** 1Centre Suisse de Recherches Scientifiques en Côte d’Ivoire, 01 BP 1303 Abidjan 01, Côte d’Ivoire; parfait.kouame@csrs.ci (P.K.K.); bassirou.bonfoh@csrs.ci (B.B.); 2Swiss Tropical and Public Health Institute, P.O. Box, CH-4002 Basel, Switzerland; georg.loss@swisstph.ch (G.L.); juerg.utzinger@swisstph.ch (J.U.); mirko.winkler@unibas.ch (M.S.W.); 3University of Basel, P.O. Box, CH-4003 Basel, Switzerland; 4Institute for Ecopreneurship, University of Applied Sciences and Arts Northwestern Switzerland, CH-4132 Muttenz, Switzerland; maryna.peter@fhnw.ch; 5Université Nangui Abrogoua, 02 BP 801 Abidjan 02, Côte d’Ivoire; wassa.diarrassouba@csrs.ci

**Keywords:** Côte d’Ivoire, drinking water, primary schools, schoolchildren, water, sanitation, hygiene (WASH)

## Abstract

In rural settings of Côte d’Ivoire, access to water, sanitation, and hygiene (WASH) at schools is often lacking. The purpose of this study was to assess the availability, quality, and use of WASH infrastructure in schools in the south-central part of Côte d’Ivoire, and to determine the hygiene practices of schoolchildren. A cross-sectional study was conducted in 20 primary schools with (n = 10) or without (n = 10) direct access to drinking water. The survey was comprised of a questionnaire directed at schoolchildren aged 8–17 years, an assessment of the WASH infrastructure, and the testing of drinking water samples for *Escherichia coli* and total coliforms. Overall, 771 schoolchildren were enrolled in the study. One out of four children (24.9%) reported that they used available toilets. Among those children not using toilets, more than half (57.5%) reported that they practised open defecation. Drinking water infrastructure was limited in most schools because of poor storage tanks, the low flow of water, or broken wells. All drinking water samples (n = 18) tested positive for total coliforms and 15 (83.3%) tested positive for *E. coli*. The lack of WASH infrastructures in primary schools in the south-central part of Côte d’Ivoire, in combination with poor hygiene practices, might govern disease transmission and absenteeism at school, especially among females.

## 1. Introduction

The major drivers of global change, including population growth, urbanization, and climate change, render the objective of reducing inequalities in access to water, sanitation, and hygiene (WASH), as stipulated in the 2030 Agenda for Sustainable Development, a major challenge [1]. In 2015, an estimated 4.5 billion people worldwide lacked access to improved sanitation, 1.8 billion used water that is likely contaminated with human excreta, and 892 million practiced open defecation [2]. These statistics are mainly driven by low- and middle-income countries (LMICs), which are particularly affected by no or only poor access to clean water and sanitation. Taken together, this resulted in 829,000 WASH-attributable deaths and 49.8 million disability-adjusted life years (DALYs) caused by diarrhoeal diseases in 2016 [3]. Sustainable development goal number 6 (SDG 6) [4], which aims to ensure the availability and sustainable management of water and sanitation for all, should remedy these adverse health issues [4].

Since most of the WASH-attributable burden of disease is caused by infections in children [5], SDG 6 is highly relevant in the context of schools [6]. Firstly, the provision of adequate WASH in schools can reduce the burden of water- and soil-related diseases [7,8,9,10,11]. For example, a reduction in infections in schoolchildren can prevent a negative impact on children’s growth, especially in relation to childhood stunting [12]. Secondly, schools are an effective platform for educating children about the role and importance of WASH for disease prevention and promoting wellbeing [10]. WASH in schools is important to reduce the risk of exposure to pathogens [13]. Thirdly, by assuring access to water and sanitation facilities, schools can serve as role model to positively influence WASH at the household level. Improving the quality of WASH facilities and their use in primary schools is essential for lowering the risk and impacts of intestinal diseases among primary schoolchildren [14].

Côte d’Ivoire, a country located along the Gulf of Guinea in West Africa, ranked at position 165 on the human development index (HDI) in 2018. Education is compulsory for children aged 6–16 years [15]. However, only 48% of children finish primary education with minimal proficiency in reading (34%). Overall, among children aged 6–16 years, 74% of girls and 79% of boys attend primary school, whereas only 35% of girls and 46% of boys attend secondary school [16]. Access to WASH infrastructures is lacking or limited, particularly in schools in rural areas. Evidence of good WASH-related practices and knowledge at schools is scarce. According to the Ministry of Planning and Development, two-thirds (63%) of the population face a high risk of exposure to faecal contamination through drinking water at the household level, and more than a fifth of the population (22%) practice open defecation [15]. UNICEF reported that half of the schools in Côte d’Ivoire do not have access to toilets and a secure water supply source [17]. The level of knowledge on WASH in schoolchildren is unknown. While there is some evidence on WASH infrastructure at the regional level, a more detailed understanding at the local level is lacking. There are no economic data on WASH expenditures for schools and, in general, at the national level in Côte d’Ivoire.

The purpose of this paper was to assess the knowledge of, and access to, WASH in primary schools in a rural part of south-central Côte d’Ivoire. More specifically, the study aimed to answer the following research questions: what is the availability of WASH infrastructure in primary schools? What is the quality of drinking water available at schools? Is WASH-related knowledge in schoolchildren associated with the availability and quality of WASH infrastructure at primary schools?

## 2. Materials and Methods

### 2.1. Study Area and Modules

A cross-sectional survey was conducted in October and November 2019 in 20 primary schools in the Taabo health and demographic surveillance systems (HDSS) [18] in the south-central part of Côte d’Ivoire (Figure 1) [19,20,21]. The study was comprised of three partially interlinked modules (Figure 2): (i) a knowledge, attitudes, and practices (KAP) questionnaire survey for schoolchildren; (ii) a WASH facility evaluation at schools with or without direct access to drinking water; and (iii) a drinking water quality assessment. The data collection process was facilitated through the involvement of the local education authorities, school principals, and the village chiefs.

At the time of the of study, a total of 48,956 individuals were registered in the Taabo HDSS. The number of children aged 6–16 years was 14,366. The water supplied for drinking and cooking in Taabo mainly originates from a groundwater source; however, surface water use for these purposes in Taabo was previously shown [19].

### 2.2. Questionnaire Survey

We employed a cross-sectional randomized cluster design for the KAP questionnaire survey (see Appendix A). For the first step, the primary schools (school performance levels 1 and 2) in the Taabo HDSS were stratified into two clusters: schools with or without direct access to drinking water (two sampling frames). For the second step, with the support of school representatives, a total of 20 school clusters were randomly selected, with 10 schools in each sampling frame. In the third step, a total of 40 schoolchildren per school (n = 20 children per school level 1 and 2) were randomly selected to participate in the study (Appendix B). For sample size calculation, the formula for normal approximation to binomial distribution without finite sample correction (level of confidence 95%; expected prevalence 50%; and precision 5%) was used. The number of children per sampling frame was calculated to be 385 children in each of the two samples, resulting in a targeted sample size of 770 schoolchildren used to identify the differences in WASH between schools with and without access to drinking water.

The survey tools were pre-tested in Taabo and adjusted to the existing context prior to its implementation. Before conducting the field activities, nine surveyors (four males and five females) were trained in conducting the surveys. For quality assurance, the local coordinator of the project supervised the field data collection, to make sure the activities were conducted in alignment with the project objectives. Collected data were checked by the team each evening. The survey was conducted at the selected schools on paper, directly in the classroom in presence of teachers.

### 2.3. WASH Facility Evaluation

WASH infrastructure and maintenance were evaluated using the qualitative Facility Evaluation Tool for WASH in Institutions (FACET) (version FACET WASH in schools, WINS version 1.3). The FACET tool is built around three core topics: (i) level of water services; (ii) level of sanitation services; and (iii) level of hygiene services. The outputs of the tool are featured in percentages, where 100% and 0% denote an optimal service and a non-existent service, respectively. The tool was adapted to the local context and covered the issue of understanding the WASH infrastructure in rural schools in the Taabo HDSS.

The implementation of FACET in primary schools was performed using Open Data Kit (ODK) software (version 1.25.1; licensed under a Creative Commons Attribution 4.0 International License) on Samsung tablets. Out of the 20 schools overall, 13 participated in the FACET survey. The number of surveys was limited to 13 as some primary schools in the Taabo HDSS were grouped at the same location, thus sharing the same water and sanitation infrastructures.

### 2.4. Drinking Water Sampling and Quality Testing

Drinking water quality was assessed according to a protocol put forth by the World Health Organization [22], which uses compact dry plates and membrane filtration for the detection of *Escherichia coli* and total coliforms (NISSUI pharma, EC 40 plates, lot 363806; Tokyo, Japan) [23]. Of note, Compact Dry *E. coli*/Coliform Count (EC) is a ready-to-use test method for the enumeration of *E. coli* and coliform bacteria. The plates are pre-sterilised and contain culture medium and a cold water-soluble gelling agent.

Drinking water samples were collected in the primary schools in Taabo with access to a water source. Two samples were collected per drinking water point, with sources including wells, SODECI water systems, pots, and storage tanks. Drinking water was collected in a sterile bag (100 mL, Nasco WHIRL-PAK). For SODECI water systems, the water was collected after having run for at least 60 s from the tap. To fill the bag with water samples from pots and storage tanks, we used a small recipient or tilted the pot if necessary. The samples were stored on ice in a cool box at 4–6 °C for a maximum of 24 h before being transferred to the Centre Suisse de Recherches Scientifiques en Côte d’Ivoire (CSRS) laboratory in Abidjan.

As a preparatory step for the sample processing, the upper parts of the filtration unit and the tweezers used were sterilised by placing them in a bowl with boiled water for 3 min. Subsequently, an ethanol wipe and sterilised tweezer were used for assembling the filtration unit. Once the unit was cooled down to an ambient temperature, 1 mL of the sample water was used to moisturise the compact dry plate. Then, the remaining 99 mL of the water sample was filtrated using a 47 mm diameter sterile membrane with a 0.45 μm pore size. Following the filtration process, the filter was placed on the compact dry plate EC and closed. The plate was then placed upside down in the incubator for 24 h at 37 °C, wrapped in a zip lock bag. After 24 h, *E. coli* and total coliform colonies were counted and recorded as colony forming units (CFU) per 100 mL. Numbers over 300 CFU/100 mL cannot be distinguished correctly and, thus, were recorded as >300 CFU/plate.

### 2.5. Statistical Analysis

The KAP survey data were recorded in EpiInfo version 3.5.1 (Centers for Disease Control and Prevention; Atlanta, GA, USA). The data were analysed using statistical software (R version 4.0.1.; Vienna, Austria). Statistical significance was defined at a 0.05 level. Descriptive statistics were performed to assess proportions and 95% confidence intervals (CIs) of the school and student characteristics, stratified by the type of school. Logistic regression analysis, adjusting for clustering effects, was used to test for differences between schools with and without access to water. To account for the disproportionate stratified sampling strategy from Taabo HDSS (sampling: 50% of schools with access to water compared to 39% schools with access to water in Taabo HDSS), all analyses were weighted using fixed weights derived from the inverse probabilities of selection. In addition, a handwashing score was used to assess the differences in handwashing practices between groups (schools with vs. schools without drinking water). The presence of good handwashing practice in four handwashing items (i.e., washing hands with soap before and after eating, after playing, and after defecation) were summed up to a score ranging from zero (never exhibiting good handwashing behaviour) to four (always washing hands in relation to the aforementioned items).

### 2.6. Ethical Considerations

Ethical clearance was obtained from the Ethics Committee Northwest and Central Switzerland (EKNZ; Req-2019-00790). In Côte d’Ivoire, because of non-invasiveness and the low risk for the participants involved, ethical approval was obtained from the institutional review board of CSRS, with minor suggestions for guardians. Written informed consent was obtained prior to the study from the schoolchildren and their guardians separately.

The student’s consent form was adapted for children and was explained in the presence of their teacher. In case of illiteracy, parents were required to sign in the presence of an impartial witness.

## 3. Results

Overall, 771 students from 20 primary schools in the Taabo HDSS participated in the survey (Table 1). Their ages ranged from 8 to 17 years. There were 394 participants (a total of 51.1%, with 47.2% at level 1 and 50.8% at level 2) from schools with direct access to water, and 377 participants (a total of 48.9%, with 50.7% at level 1 and 49.3% at level 2) from schools without direct access to water. In terms of gender, there were slightly more females than males participating in the survey (access to water: 52.8% females; no access to water: 55.7% females).

### 3.1. WASH-Related KAP in Schoolchildren

School toilets or latrines were used by one quarter of the children (24.9%; 95% CI: 21.9–28.1%) (Table 2). No statistically significant difference in toilet use could be found between schools with and without access to drinking water. After excluding schools identified as not having any toilet facilities, only 28.2% (95% CI: 24.8–31.8%) of the remaining schoolchildren used the toilets at schools. Despite the FACET survey classifying the presence of latrines or toilets at certain schools, some students stated that their school did not have these facilities. One of these schools, Aheremou 1, only had toilets for female students. Further, two schools, N’Denou 1 and 2, did have toilet facilities, but they equated to one toilet per 70–80 students. The two main reasons students did not use the school toilets related to bad smell (14.7%; 95% CI: 10.9–18.5%) and uncleanliness (21.8%; 95% CI: 18.0–25.7%). In addition, when the participants were asked whether the toilets were always open, only one third (36.7%; 95% CI: 32.8–40.6%) responded positive. The children who did not use the toilets at school reported that they used toilets at home (15.3%; 95% CI: 12.1–18.6%), practiced open defecation (57.5%, 95% CI 54.0–61.2%), or used the area behind the latrines (2.2%, 95% CI: 0.0–5.0%). Due to the sensitive nature of this question, 16.3% (95% CI: 13.9–19.0%) of the students did not answer.

As shown in Table 3, about a third of the students did not wash their hands with soap before eating (28.0%; 95% CI: 24.9–31.4%) or after defecation (37.0%; 95% CI: 33.5–40.5%). There were no significant differences in these behaviours with regard to whether or not the children had access to drinking water at school. When asked on which occasions hands should be washed with soap, 80.0% (95% CI: 77.2–82.7%) of the schoolchildren said before eating and 75.4% (95% CI: 72.4–78.5%) mentioned after defecation. The reasons mentioned for handwashing included to avoid becoming sick (39.4%; 95% CI: 35.8–43.3%) or having microbes on hands (16.2%; 95% CI: 12.6–20.0%). Furthermore, handwashing was perceived as removing dirt (18.5%; 95% CI: 14.9–22.4%) and bad smells (1.7%; 95% CI: 0.0–5.5%). The self-reported handwashing score, which was a maximum of four and consisted of washing the hands before and after eating, after playing, and after defecation, was achieved by a quarter of the surveyed children (25.0%; 95% CI: 21.4–28.7%). A total of 17.0% (95% CI: 13.4–20.7%) of students reported they did not wash their hands in the aforementioned situations, whereas 7.4% (95% CI: 3.8–11.1%) believed that they did not have to wash their hands in those situations. The mean of the self-reported handwashing score in practice (mean = 2.25) was significantly lower compared to the mean of the handwashing knowledge score (mean = 2.78) (V = 2.22, *p*-value < 0.05). No statistically significant difference between students from schools with or without access to drinking water could be detected in the good handwashing behaviour practice and knowledge scores.

When asked whether water can cause diseases, three quarters (75.9%; 95% CI: 73.0–79.0%) of the students responded yes. Significantly more students from schools without access to water stated that water can cause diseases (84.1%; 95% CI: 80.6–87.8%; χ^2^ = 32.2, *p*-value < 0.001) compared to students from schools with access to water (68.0%; 95% CI: 63.5–72.8%). Most of the diseases identified by the students were of gastro-intestinal in nature (Table 4). Additionally, other diseases, such as malaria, skin diseases, or HIV/AIDS, were provided in response. More than a third of the students (35.9%; 95% CI: 32.6–40.0%) did not answer the question about potential water-related diseases that may be contracted with limited WASH access. The students reported that they were exposed to freshwater sources, such as rivers or lakes, while doing the laundry (46.3%; 95% CI: 42.7–50.1%), playing (33.1%; 95% CI: 29.7–36.6%), fishing (15.9%; 95% CI: 12.8–19.1%), and working (12.6%; CI: 9.1–16.2%).

### 3.2. Availability and Quality of WASH Infrastructures

Among the 20 participating schools, seven were grouped at the same location and shared WASH infrastructure (Appendix B), resulting in a sample of nine school groups that were evaluated with FACET and water sampling and 13 school groups for which the statistical analysis was adjusted. The survey was conducted with the principal in all the schools. Major WASH-related issues identified by the respective principals were: (i) not having support for the construction of additional latrines at the schools; (ii) not having water filters in each classroom; (iii) the need to increase the number of taps (two to three per school) and the number of latrines; (iv) not having handwashing materials, such as soap; (v) the need for training to learn waste recycling at schools; and (vi) having non-operational toilets. In N’Denou, the consumption of unsafe well water by teachers and students was mentioned as an additional issue. The average number of children per school was 227 (range: 176–304) with an average of 113 (range: 86–156) females and an average of 114 (range: 82–149) males. Almost all the schools employed six teachers, with the exception of two schools, which employed seven teachers. The drinking water infrastructure of 11 schools was considered to be inadequate because of the low flow of water, the absence of infrastructure, or because the water source was located outside of the school. Only two of the schools had basic water infrastructure, which mainly consisted of tap water from a SODECI pipe. On average, the schools had 1.3 water access points (range: 0–4 water access points per school).

Regarding water quality evaluation, school authorities noted that the water access points were not tested to assess contamination with faecal indicator organisms. In terms of sanitation, the survey showed that two schools (15.4%, n = 13) did not have any toilets, eight schools (61.5%) had pit latrines, and three schools (23.1%) had flush toilets. In addition, four schools offered gender-separated toilets. None of the toilet facilities were lit. Regarding the level of sanitation, four schools had access to basic sanitation, seven had limited access, and two did not have access to toilets at all. When looking at hand hygiene infrastructure, only one school had an available handwashing station, but there was no soap, and it was unusable because no water was available. Furthermore, all the schools burned and/or dumped their solid waste openly on the school premises.

### 3.3. Drinking Water Quality

The results showed that out of 18 drinking water samples, 15 (83.3%) samples tested positive for both *E. coli* and total coliforms in the selected primary schools of the Taabo HDSS. In two sample sites in N’Denou 1 and Léléblé 3 and 4, concentrations of *E. coli* and total coliform of >300 CFU/100 mL were found (Figure 3). The concentration of *E. coli* varied from 4 to 6 CFU/100 mL in Kotiessou, while in the other sample sites, the drinking water was not contaminated by *E. coli*. In terms of total coliforms, only one sample site with two schools (Taabo Village 1 and 2) had no drinking water contamination. The levels of total coliform contamination varied from 12 CFU/100 mL to >300 CFU/100 mL among the contaminated samples. The two sample sites of N’Denou 1 and Léléblé 3 and 4 had the highest concentration of total coliforms, followed by Ahondo 1 and 2 (152–200 CFU/100 mL), and Kotiessou 1 and 2 (17–22 CFU/100 mL).

## 4. Discussion

This study assessed the availability, quality, and use of WASH infrastructure in the context of primary schools in the Taabo HDSS in the south-central part of Côte d’Ivoire. We found poor WASH-related practices, major shortcomings in the availability of WASH infrastructure, and poor microbiological quality of the drinking water available in the sampled primary schools. To the best of our knowledge, this is the first investigation to provide reliable data on WASH-relevant knowledge and behaviour among schoolchildren in the Taabo HDSS, which will facilitate planning for sustainable WASH interventions at primary schools.

### 4.1. WASH-Related Practices

Only a third of students from schools with toilets consistently used these facilities. Reasons for not using sanitation facilities included the perceived absence of hygiene, a lack of access for certain groups (e.g., females), and the unavailability of water. Most of the students knew when and why they are supposed to wash their hands with soap. However, despite this knowledge, about a third of the schoolchildren reported that they did not wash their hands before eating and after defecation. This discrepancy of knowledge and practice might arise due to the lack of handwashing facilities or the unavailability of clean water. It follows that open defecation practices were reported by more than half of the students who did not use sanitation facilities at school. These practices contribute to the transmission of infectious diseases such as diarrhoea, hookworm, trachoma [7,24,25], schistosomiasis [26], and intestinal protozoa infections [27]. Additionally, open defecation reduces the privacy of children, especially girls, and can put them at a higher risk of sexual exploitation [28]. Furthermore, open defecation affects the children’s well-being through a lack of privacy. For example, as observed in schools in the Taabo HDSS, and highlighted by relevant studies in other settings, girls who lack the privacy to maintain their menstrual hygiene might miss school during this time of the month [13,29,30].

Good hand hygiene practices, including the use of soap, contributes to the reduction of a variety of diseases [7,31]. However, the availability of soap was very limited in Taabo. Most of the students in our study reported that they washed their hands to remove microbes, dirt, and bad smells. Similarly, a lack of soap for handwashing practices was reported in other rural settings in Africa [32,33]. Soap provision at the school level is particularly relevant in settings where open defecation is common, as is the case in the primary schools in the Taabo HDSS, since this increases the risk of contaminated hands and, thus, disease transmission. Hence, to assure the health and well-being of primary schoolchildren in Taabo and to minimise absenteeism due to diseases, there is a pressing need to improve handwashing facilities and access to adequate sanitation [20,34].

### 4.2. Availability and Quality of WASH Infrastructure

Our study showed that even if toilets were available at schools, there were often too few toilets for the number of children, they were not accessible because of poor maintenance, or they were only accessible to teachers or to a fraction of the schoolchildren. The observed shortcomings in the availability of toilets at most schools in Taabo went hand-in-hand with open defecation practices. The availability of toilets or latrines and the usability when available are important for the wellbeing of children, as they contribute to a reduction in preventable, water-related morbidity and mortality. In addition to poor availability and access to toilets, drinking water infrastructure in most schools in Taabo was either not available or inadequate. This was confirmed by drinking water samples: four out of five samples from wells, hand-pumps, pots, and storage tanks were contaminated with *E. coli*, which is an indicator of faecal contamination [35,36]. Apart from schools, poor sanitation infrastructure and drinking water sources were reported in households of the study area by Schmidlin et al. [19]. The authors found that in 71% of participating households, toilet facilities were lacking and that defecation practices were significantly associated with parasitic infection.

Studies in Ghana and Indonesia showed that safe WASH infrastructures at schools increased children’s sense of privacy, while this added confidence was shown to reduce absenteeism at school [37,38,39]. Hence, by making the provision of adequate WASH infrastructure a priority, including well maintained toilets, safe drinking water, and appropriate handwashing facilities, there is an opportunity for the education and health sectors in Côte d’Ivoire to jointly promote health and well-being in schoolchildren through multi-sectoral action, which is considered essential for the achievement of the 2030 Agenda for Sustainable Development [40].

### 4.3. Quality of Drinking Water

Drinking water quality in facilities provided at schools in Taabo was poor, as almost all water systems contained *E. coli* and total coliforms. A potential factor contributing to the poor quality of drinking water in the study area was the unhygienic storage of water in tanks and containers, such as pots managed by children. These practices are known as risk factors for the presence of pathogens in drinking water [41]. In addition to poor storage, drinking water can be contaminated by poor sanitation infrastructure and poor waste management [42,43]. Both problems were observed in the FACET evaluation.

Poor drinking water quality is a known problem in LMICs from different geographical contexts [42,43]. For example, in Nepal, a quantitative microbial risk assessment (QMRA) was conducted for evaluating the risk of diarrhoeal diseases caused by faecal contamination in several water sources [44]. This study found that the water used in households, including water for drinking and washing, had higher risks associated with it than water from the water source. In India, for example, drinking water from borewells and hand-pumps was contaminated by thermo-tolerant coliforms leading to diarrhoeal diseases [45]. Considering the potential burden of disease, which could be prevented in LMICs with improved access to drinking water, infrastructural and behavioural interventions are urgently needed [27].

### 4.4. Strengths and Limitations

This study has several limitations. Firstly, due to the cross-sectional design, no causal associations could be presented. However, our study was explorative and should serve as a basis for later studies with a more sophisticated epidemiological design. Secondly, the survey with the students relied on self-reporting and was therefore prone to recall bias. Additionally, questions about toilet use and defecation behaviour are regarded as sensitive by many and might underlie a social desirability bias. To circumvent these potential biases in future studies, objective observations of handwashing behaviour or qualitative data collection are indicated. Thirdly, reasons for not using the school’s toilet facilities varied greatly in the survey. Focus group discussions might be helpful to determine the most important factors for a sanitation facility to be usable and appropriate to their needs. Additionally, the school’s age should be recorded to better address problems with infrastructure maintenance. Shortcomings observed in this study about poor drinking water infrastructure, toilets, and handwashing stations showed that SDG 6 is not well addressed nor close to being achieved in schools of the Taabo HDSS. Finally, this study is representative for the Taabo HDSS area in the south-central part of Côte d’Ivoire. In our sampling strategy, we oversampled the schools with access to water to better ensure appropriate sample size when associating access to water with student knowledge and practices and our results were weighted accordingly. However, the results have limited generalisability to Côte d’Ivoire or the African continent, as social and cultural practices differ greatly. This limitation is not restricted to our study, but to the whole Taabo HDSS population, since 85.6% is primarily Akan in ethnicity, which is different from, for example, the ethnicity of the Senoufos residing in the North of the country [18,46]. Nevertheless, the findings are in line with other studies from LMICs [39,47]. Despite these limitations, our study is an important contribution and identified gaps that need to be addressed in order to achieve SDG 6 in rural Côte d’Ivoire by 2030. Our findings also imply that more sustainable solutions for infrastructural and behavioural interventions need to be implemented to guarantee the long-term success of these goals.

## 5. Conclusions

This study showed that the drinking water in Taabo schools did not meet the quality standards set by the WHO [22] and the sanitation infrastructure was, to a large extent, lacking or inadequate. Moreover, we did not find any access to hygiene infrastructure, such as handwashing stations with soap or menstrual hygiene management facilities, in the primary schools of Taabo. Therefore, SDG target 6.2, which stipulates the achievement of “access to adequate and equitable sanitation and hygiene for all and end open defecation, paying special attention to the needs of women and girls, and those in vulnerable situations”, will be difficult to reach by 2030 in the Taabo HDSS. To improve the situation, an entry point could be the incorporation of the topic of WASH, and its relation to health and well-being, into the curricula of rural primary schools in Côte d’Ivoire. This will not only lead to improved hygiene practices among schoolchildren, but also promote awareness in teachers and schools about the importance of functional WASH infrastructure. Such an initiative could be linked to targeted investments in WASH infrastructure in schools, while paying particular attention to the personal hygiene needs of females to bridge potential gender gaps. Apart from initiatives that aim to improve WASH at schools, future research on this topic is required. To address specific WASH-related issues and the needs of schoolchildren with a gendered perspective, focus group discussions with students and teachers are recommended. Moreover, qualitative research would be useful to deepen the understanding of cultural and social issues linked to WASH. With the combination of these approaches and quantitative WASH studies in schools of Côte d’Ivoire, and sub-Saharan Africa more broadly, comprehensive and inclusive WASH assessments would become a reality.

## References

## Figures and Tables

**Figure 1 ijerph-18-08863-f001:**
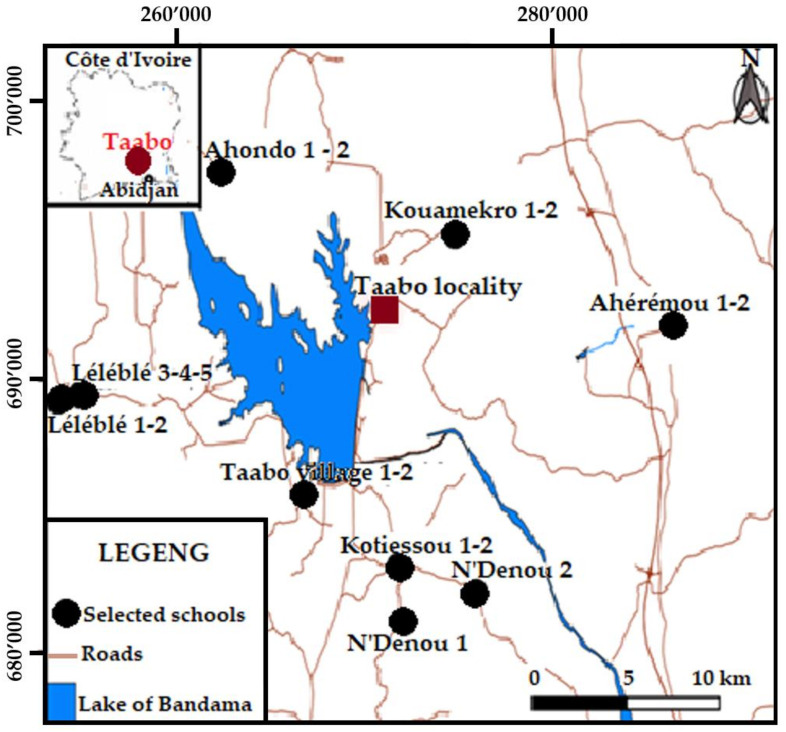
Study area and location of schools in Taabo health and demographic surveillance system in the south-central part of Côte d’Ivoire.

**Figure 2 ijerph-18-08863-f002:**
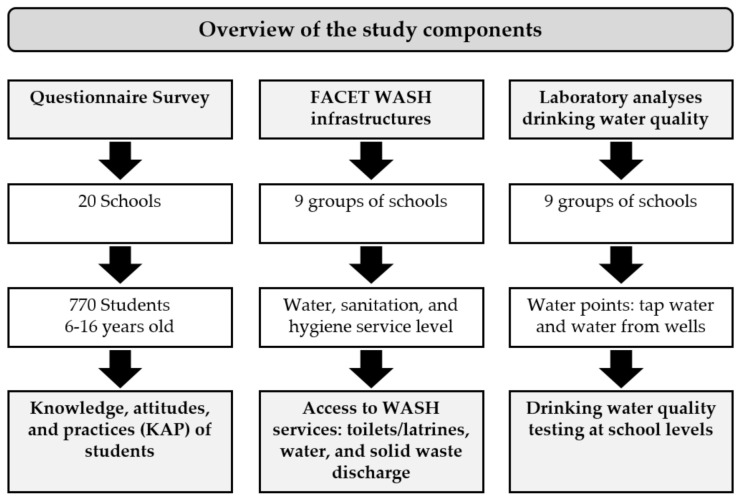
Research framework, showing the study method components.

**Figure 3 ijerph-18-08863-f003:**
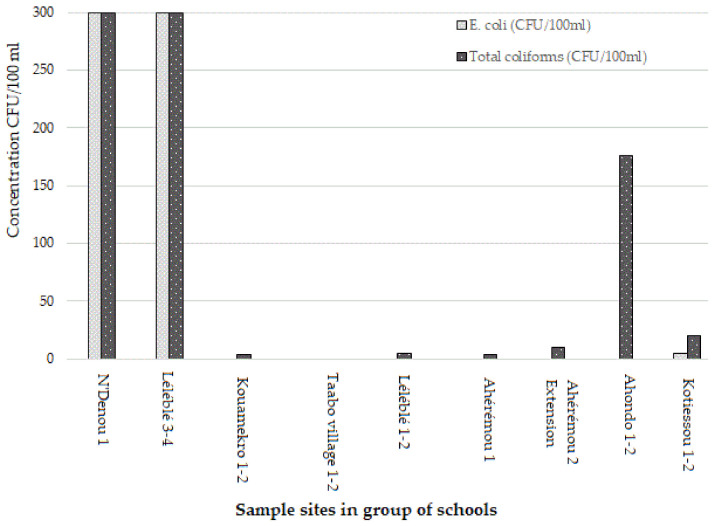
Drinking water quality depicted in colony forming units (CFUs) per 100 mL of *E. coli* and total coliforms in primary schools of the Taabo HDSS (October–November 2020).

**Table 1 ijerph-18-08863-t001:** Participant characteristics by access to drinking water^1^.

	Schools with Access to Drinking Water (95% CI)	Schools without Access to Drinking Water (95% CI)	*p*-Value for Difference
**Number of students** Female (%)	394 52.8 (47.7–57.9)	377 55.7 (50.7–61.0)	0.382
**Age** (median in years) **IQR**	11 2	11 2	0.847
**School level (%)** Level 1 (CM1) Level 2 (CM2) NA	47.2 (42.1–52.4) 50.8 (45.7–55.9) 2.0 (0.0–7.2)	50.7 (45.6–56.1) 49.3 (44.3–54.7)	0.157

^1^ Statistical tests were conducted with logistic regression, adjusted for clustering, at school group level and weighted to avoid oversampling of schools with access to water (each variable was tested for presence vs. absence unless stated otherwise). CI, confidence interval; IQR, interquartile range.

**Table 2 ijerph-18-08863-t002:** Toilet/latrine related knowledge and practices ^1^.

	Schools with Access to Drinking Water	Schools without Access to Drinking Water	*p*-Value for Difference
**Toilet/latrine use at school (%)**	27.7 (23.4–32.2)	22.0 (18.8–26.3)	0.530
**Reasons for not using the toilet/latrine (%)** Bad smell Dirty Not functioning Not enough toilets No privacy No water No soap Toilets are always open	16.5 (11.9–21.6) 27.9 (22.8–33.3) 6.1 (2.0–10.6) 1.5 (0.0–5.8) 1.0 (0.0–5.2) 7.4 (3.0–11.8) 7.6 (3.3–12.1) 31.5 (26.4–36.9)	12.7 (8.0–17.9) 15.4 (10.6–20.5) 3.7 (0.0–9.0) 1.1 (0.0–6.3) 0.3 (0.0–5.5) 8.5 (3.7–13.7) 7.9 (3.2–13.2) 39.0 (33.7–44.7)	0.309 0.736 0.747 0.682 0.619 0.098 0.229 0.263
**Preferred place of** **defecation/urination while at school (%):** Toilet/latrine In the bushes Behind the latrine At home	4.1 (0.8–7.8) 56.9 (52.0–62.1) 3.0 (0.0–6.6) 15.7 (11.4–20.2)	6.6 (2.4–11.2) 58.1 (53.1–63.3) 1.3 (0.0–5.5) 14.9 (15.4–25.0)	0.384 0.355 0.212 0.916

^1^ Statistical tests were conducted with logistic regression adjusted for clustering at school group level and weighted to avoid oversampling of schools with access to water (each variable was tested for presence vs. absence unless stated otherwise). Values in brackets are 95% confidence intervals.

**Table 3 ijerph-18-08863-t003:** Handwashing and water-related knowledge and practices ^1^.

	Schools with Access to Drinking Water	Schools without Access to Drinking Water	*p*-Value for Difference^2^
**Handwashing with soap/detergent** **(practice) (%)** Before eating After eating After defecation After playing	70.3 (66.0–75.1) 58.1 (53.3–63.3) 60.2 (55.3–65.3) 38.6 (33.8–43.6)	72.7 (68.4–77.4) 49.1 (44.0–54.5) 65.0 (60.2–70.0) 35.5 (30.8–40.7)	0.668 0.224 0.421 0.540
**Handwashing with soap/detergent (knowledge) (%)** Before eating After eating After defecation After playing	78.9 (75.1–83.0) 69.8 (65.5–74.6) 73.1 (68.8–77.5) 62.9 (58.1–67.9)	80.9 (77.2–84.9) 59.4 (54.4–64.6) 77.7 (73.7–82.0) 52.5 (47.5–57.9)	0.705 0.145 0.475 0.100
**Handwashing score practice (comparison of means)**	2.27	2.22	0.794
**Handwashing score knowledge** **(comparison of means)**	2.85	2.71	0.479
Water can cause diseases (%)	68.0 (63.5–72.8)	84.1 (80.6–87.6)	<0.001 ***
**Fresh water exposure (%)** Doing the laundry Playing Fishing Working	55.8 (51.0–61.2) 32.0 (27.4–36.9) 20.1 (15.7–24.7) 19.8 (14.7–24.6)	36.3 (31.3–41.6) 34.2 (29.4–39.3) 11.7 (7.4–16.0) 5.0 (0.3–10.3)	<0.001 *** 0.773 0.075 <0.001 ***

^1^ Statistical tests were conducted with logistic regression adjusted for clustering at school group level and weighted to avoid oversampling of schools with access to water (each variable was tested for presence vs. absence). ^2^ Significance code: * = *p*-value < 0.05, ** = *p*-value < 0.01, and *** = *p*-value < 0.001. Values in brackets are 95% confidence intervals.

**Table 4 ijerph-18-08863-t004:** The student’s answers to the question of which diseases are caused in relation to lack of water or contaminated water.

Disease	Number of Students	%
Stomach ache	326	42.3
Diarrhoea	126	16.3
Typhoid fever	13	1.7
Cholera	14	1.8
Vomiting	9	1.2
Schistosomiasis	16	2.1
Malaria	127	16.5
Headaches	51	6.6
Skin rashes	26	3.4
AIDS or HIV	24	3.1
Fever	18	2.3
Diabetes	7	0.9
Not applicable. or do not know	277	35.9

## Data Availability

The data that support the findings of this study are available from the corresponding author. Restrictions apply to the availability of these data, which were used under license for this study. Data are available from the authors with the permission of Swiss TPH in Switzerland and CSRS.

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
