# Peer review of "Access to Water and Sanitation Infrastructures for Primary Schoolchildren in the South-Central Part of Côte d’Ivoire"

_ijerph, 2021, doi:10.3390/ijerph18168863_

Round 1
Reviewer 1 Report
While the study obviously aims to identify differences between schools with and without WASH facilities these differences do not translate into burden of diseases differences, as vaguely suggested in the introduction. The introduction could be strengthened with some economic data, should they be readily available. For example, How much Côte d'Ivoire government funding is allocated specifically to enhance the installation of WASH facilities in schools? Is there a budget allocation in the budgets of the health or education sectors (or both) for the operation and maintenance of WASH facilities in schools? Has any funding been allocated to the development of specific WASH modules to be included in the primary and secondary school curricula?
The study design and the sampling approach are solid. However, the statement in line 252, seven of the 20 participating schools were grouped at the same location and shared WASH infrastructure undermines the reader's appreciation of the randomness of the sample - couldn't such a cluster have been avoided? Can the adjustments to avoid a possible bias be explained better?
Looking at the three study components, one wonders why the teachers (6 or 7 per school it says in line 257) were not included in a separate survey. That would have added valuable information at little extra costs, about education staff's attitudes and perceptions, and insights what the education sector staff sees as obstacles and opportunities.
In the water quality sampling it is standard practice to let the tap run for say 5 or 10 seconds before taking the sample, but 60 seconds seems far beyond that and also far removed from reality. Why not take multiple samples at time intervals?
In line 311 and further menstrual hygiene is mentioned - how relevant is that for primary school girls - at what age does the menstrual cycle get started in girls in Côte d'Ivoire?
Two language editing issues: in line 360: "was", not "were", and in reference 15 a typo: "development" not develoment".
Reviewer 2 Report
3. Results
On the footnotes of the Tables in the Results section, add a Key footnote which explains the degrees of significance for the asterisks, e.g. *** =p-value <0.001. This must done for all tables where the asterisks are indicated.
5. Conclusions
Areas for future research directions must be added at the end of this section.
Reviewer 3 Report
The research is relevant however the results have limited generalizability regarding Coet d Ivoire or the African continent as social and cultural practices differ. Also, focus groups would have been a great method to use for your research which would have provided rich details on access to water and sanitation infrastructures for primary schoolchildren in the South-central part of Cote d'Ivoire. Finally, the lack of water and sanitation facilities for the school children impact their educational process in the schools.
Reviewer 4 Report
Access to clean water and sanitation for all is an important part of Sustainable Development Goals. However, it remains a serious challenge for many low- and middle-income countries. This paper focuses on primary schoolchildren in the south-central part of Côte d’Ivoire, and its methods and findings are impressive.
Comment 1
This paper discussed several important problems, including the availability of WASH infrastructures, the quality of drinking water available and the correlation between WASH-knowledge and availability and quality of WASH infrastructure in primary schools in the south-central part of Côte d’Ivoire. Both research topic and its questions were very clear and concrete.
Comment 2
It is necessary to clarify the internal logical connection between sanitation facilities, hygiene practices and disease. By constructing the research framework, this paper presented a clear thinking and valid methods, and its conclusions were explicit. It basically realized the purpose of research, which contributes to better understand the key of health inequalities and solve realistic problems.
Comment 3
It is laudable that a gender perspective was clearly visible in this paper. What are the main reasons of the gender differences? Crucially, how to bridge the gender gap presented in this study? I think it is an important topic worth exploring. It is expected that authors will deepen the relevant research in the future.
